# Porous Nanomaterials Targeting Autophagy in Bone Regeneration

**DOI:** 10.3390/pharmaceutics13101572

**Published:** 2021-09-28

**Authors:** Qing Zhang, Lan Xiao, Yin Xiao

**Affiliations:** 1Affiliated Stomatology Hospital of Guangzhou Medical University, Guangzhou Key Laboratory of Basic and Applied Research of Oral Regenerative Medicine, Guangzhou 510182, China; zhangqing@gzhmu.edu.cn (Q.Z.); l5.xiao@qut.edu.au (L.X.); 2Laboratory for Myology, Department of Human Movement Sciences, Faculty of Behavioural and Movement Sciences, Vrije Universiteit Amsterdam, Amsterdam Movement Sciences, 1081 BT Amsterdam, The Netherlands; 3Centre for Biomedical Technologies, School of Mechanical, Medical and Process Engineering, Queensland University of Technology, Brisbane, QLD 4000, Australia; 4The Australia-China Centre for Tissue Engineering and Regenerative Medicine (ACCTERM), Queensland University of Technology, Brisbane, QLD 4000, Australia

**Keywords:** nanomaterials, autophagy, osteogenesis, immune microenvironment, bone, regenerative medicine

## Abstract

Porous nanomaterials (PNMs) are nanosized materials with specially designed porous structures that have been widely used in the bone tissue engineering field due to the fact of their excellent physical and chemical properties such as high porosity, high specific surface area, and ideal biodegradability. Currently, PNMs are mainly used in the following four aspects: (1) as an excellent cargo to deliver bone regenerative growth factors/drugs; (2) as a fluorescent material to trace cell differentiation and bone formation; (3) as a raw material to synthesize or modify tissue engineering scaffolds; (4) as a bio-active substance to regulate cell behavior. Recent advances in the interaction between nanomaterials and cells have revealed that autophagy, a cellular survival mechanism that regulates intracellular activity by degrading/recycling intracellular metabolites, providing energy/nutrients, clearing protein aggregates, destroying organelles, and destroying intracellular pathogens, is associated with the phagocytosis and clearance of nanomaterials as well as material-induced cell differentiation and stress. Autophagy regulates bone remodeling balance via directly participating in the differentiation of osteoclasts and osteoblasts. Moreover, autophagy can regulate bone regeneration by modulating immune cell response, thereby modulating the osteogenic microenvironment. Therefore, autophagy may serve as an effective target for nanomaterials to facilitate the bone regeneration process. Increasingly, studies have shown that PNMs can modulate autophagy to regulate bone regeneration in recent years. This paper summarizes the current advances on the main application of PNMs in bone regeneration, the critical role of autophagy in bone regeneration, and the mechanism of PNMs regulating bone regeneration by targeting autophagy.

## 1. Introduction

Bone is a metabolically active tissue that maintains physiological function and homeostasis through a continuous remodeling process that consists of bone resorption and formation [1]. For large bone defects caused by tumor, trauma, inflammation, or infection, it is necessary to implant materials to promote new bone regeneration and restore bone function. From the view of materials, healthy bone is a complex natural material composed of organic nanomaterials (collagen, nanofibers) and inorganic nanomaterials (nano-hydroxyapatite) with a multi-level structure from the micro-nanometer to the macro-level [2]. This multilayered structure is also linked to responding to stimuli/injury and activating regeneration [2]. Therefore, nanomaterials have a broad prospect for the development of functional bone regenerative materials. In recent years, nanomaterials have attracted increasing interest in bone regeneration studies, and many reviews have reported on the application of nanomaterials for bone regeneration. For instance, as early as 2009, Webster, T.J., reviewed the prospects for nanomaterials in bone, cartilage, blood vessel, nerve, and bladder tissue engineering [3]. Srinivasan, D.K., elucidated the use of nanoparticles as a drug delivery system to improve bone regeneration [4]. Liu, C., summarized the development of nanomaterials that can promote bone regeneration in a mimicked bone-healing model (e.g., compositional, nanocrystal formation, structural, and growth factor-related mimicking) [2]. Meanwhile, the mechanisms by which nanomaterials regulate cell behavior have been widely investigated [5]. Studies on the interactions between nanomaterials and cells have shown that cell phagocytosis and clearance of nanomaterials, cell function maintenance, cell differentiation, and stress response are strictly regulated by autophagy [6].

Autophagy is the response of cells to stress. It is an evolutionarily conserved process with multiple roles. Primarily, it maintains intracellular homeostasis by degrading and circulating metabolites within cells, providing energy and nutrients, eliminating cytotoxic substances such as damaged proteins and organelles [7]. Interestingly, in the skeletal system, autophagy activated by specific nanomaterials contributes to osteogenic differentiation [7,8]. At the same time, other studies have shown that the toxic effects of nanomaterials may also be associated with autophagy [9], which leads to bone loss [10] and osteolysis [11,12]. During bone remodeling, autophagy plays a vital role in the differentiation of osteoclasts and osteoblasts via the mediating immune regulation [13]. These results suggest that autophagy plays a bi-directionally regulatory role in the process of promoting or inhibiting osteogenesis; therefore, targeting autophagy is of great significance for the design of bone regenerative nanomaterials. Hence, targeting autophagy may be a practical approach for promoting bone regeneration.

PNMs are nanomaterials with a porous structure and high surface ratio, which are widely used in the fields of biomedical engineering [14,15,16,17,18] such as bone regeneration [18,19,20,21], drug delivery [22,23,24], cell trace, and regulation of cell differentiation. Since 2014, it has been reported that mesoporous bioactive glass nanomaterials can promote osteogenic differentiation through activation of autophagy [25]. Targeting autophagy has become a new research focus in the application of PNMs for bone regeneration. In the current review, the main application of PNMS in bone regeneration is summarized. Then, the crucial regulatory role of autophagy in bone regeneration is briefly introduced. The regulatory roles of PNMs, including mesoporous silica nanoparticles (MSNs) [20,26,27], mesoporous hydroxyapatite nanoparticles (HAP) [28,29,30], alumina nanoparticles (Al_2_O_3_) [31,32], mesoporous bioactive glass nanoparticles (MBGNs) [33,34,35,36], mesoporous ceria (MCeO_2_) [37,38], and metallic oxides [39,40] in bone regeneration via targeting autophagy, are reviewed and discussed.

## 2. PNMs for Bone Regeneration

The type, structure, and morphology of nanomaterials have been continuously improved to achieve better bone regeneration. In particular, PNMs with porous structures have aroused the interest of researchers. Compared with nanomaterials without pores, porous materials have higher porosity and higher specific surface area. The high porosity facilitates the design for an excellent drug delivery carrier, and the high specific surface area makes it easily modifiable with bioactive molecules. The application of PNMs in bone regeneration is mainly focused on the following four aspects (Figure 1).

(1) PNMs can be designed as efficient nanocarriers to promote bone regeneration through (controlled) delivery of beneficial factors such as small molecule compounds, genes, and proteins. A variety of microporous (<2 nm), mesoporous (2~50 nm), and macropore nanomaterials (>50 nm) are designed as drug delivery carriers [41]. In particular, due to the high porosity and specific surface area, PNMs are widely used to deliver drugs for bone regeneration such as mesoporous silica nanoparticles [42,43,44] and mesoporous hydroxyapatite nanoparticles [29,30,45]. The hollow and mesoporous structure of nanomaterials can enhance drug loading efficiency [46]. For example, Hae-Won Kim [36] used hollow porous nanoparticles to deliver small genetic molecules to silence the target gene thereby, in turn, stimulating osteogenic differentiation. In another study, mesoporous silica nanoparticles have been combined with hydroxyapatite to generate a composite coating for implant surface modification, which served as a drug-delivery tool to suppress osteoclastogenesis for improving bone regeneration and osteointegration [27]. These nanocarriers are often combined with other materials, such as polymers [47], hydrogels [48], and metal materials [49], to promote bone regrowth;

(2) PNMs can be used as an imaging contrast agent to trace the cells and monitor real-time tissue regeneration. Stem cell-based therapy is a promising approach in regenerative medicine [50]. However, the distribution and migration of stem cells after transplantation cannot be effectively monitored in vivo. To achieve this goal, an indirect or direct cell tracker is preferred [51]. Current approaches, such as indirect fluorescent reporter gene labeling, are challenging to obtain deep structure images in vivo with limited detection methods; furthermore, transgenic cells are difficult to use in clinical treatment due to the regulatory issues [52]. On the other hand, nanomaterials can be used to directly label cells and can be simultaneously detected with a variety of imaging methods including magnetic resonance (MRI) [53], computed tomography (CT) [54], and photoacoustic imaging (PI) [55,56]. Some products are already commercialized for this purpose as reviewed by Wang et al. [57]. PNMs have performed well in cell imaging studies. For instance, mesoporous silica shows excellent potential in stem cell tracking. The synthesized PEGylated gold/silica nanoparticle can simultaneously be detected by MRI, CT, and fluorescence imaging (FI) [58]. Jokerst, J.V., developed exosome-like silica nanoparticles, serving as a novel ultrasound contrast agent for stem cell imaging [59]. Gadolinium^3+^-doped mesoporous silica nanoparticles also served as a potential magnetic resonance tracer for monitoring the migration of stem cells in vivo [60];

(3) PNMs have been used to fabricate or modify tissue-engineering scaffolds. The primary goal of tissue-engineered scaffolds is to develop an implant to replace the original bone tissue while supporting the regeneration process [61]. As mentioned above, natural bone has a micro-nanometer to a macroscopic hierarchical structure; thus, the tissue-engineered scaffolds must be designed in a three-dimensional structure with a highly porous feature, forming an interconnected pore network to mimic the structure of natural bone [62]. To achieve this goal, nanomaterials have been used to develop tissue-engineering scaffolds to improve the bone formation properties, such as cell growth, nutrient transport, new bone growth, and angiogenesis [52]. The physicochemical stability and mechanical properties of collagen hydrogel were improved by modification of aminated mesoporous bioactive glass in order to be better applied in tissue engineering stem cell culture [33]. Titanium dioxide nanotubes (TiO_2_ NTs) are a type of classic PNMs with a diameter in the range of 30–100 nm, which have been widely used to construct and modify scaffolds to enhance cell attachment [63] and osseointegration [64,65]. N.K. summarized the potential applications of TNTs in implants [66]. Compared with the untreated titanium, TiO_2_ NT-modified titanium enhanced the deposition of type I collagen when implanted into the porcine frontal skull. In addition, bone implants with TiO_2_ NT modification have good contact with bone and will not be damaged due to the fact of simple stress [67]. In a tibial bone defect model of rabbits, the TiO_2_ NT-modified implants induced a nine-fold increase in the bone binding rate compared to the non-modified implants [68]. In vitro and in vivo studies have shown that titanium dioxide nanotubes can increase the deposition of calcium and phosphorus and enhance the expression of osteogenesis genes such as alkaline phosphatase (ALP), osterix (Osx), and collagen-I (COL-I) [69];

(4) PNMs have been regarded as an active substance to directly modulate cell behavior (cell adhesion and differentiation). Nanomaterials, themselves, are excellent enhancers of new bone formation. It is known that gold nanoparticle size and shape can influence the osteogenesis of mesenchymal stem cells [70]. In addition, porous materials, such as mesoporous bioactive glass nanoparticles [33,34,35,36,71], mesoporous hydroxyapatite [72], and mesoporous ceria (MCeO2) [37,38], have been found to directly promote osteoblast differentiation. For instance, cerium oxide nanoparticles-modified bioglass could enhance bone regeneration by activating the extracellular signal-regulated kinase (ERK) signaling pathway [38]. In another study, ceria nanocrystals, decorated with mesoporous silica nanoparticles, have been found to facilitate tissue regeneration via inducing reactive oxygen species-scavenging (therefore, avoiding tissue damage and inflammation), suggesting its potential in bone regeneration [73] Mesoporous Ce-doped bioactive glass nanoparticles could improve osteogenesis via Ce-induced anti-oxidation and anti-inflammation [74]. Similarly, nanoceria encapsulated within mesoporous silica nanoparticles (Ce@MSNs) have been found to facilitate bone regeneration in osteoporosis and that nanoceria could induce anti-oxidation and facilitate osteogenesis, ref. [75] suggesting that ceria could be considered a critical component in bone regenerative PNMs design. Moreover, ionic-doped PNMs can promote osteogenic differentiation and facilitate angiogenesis [76] and regulate immunity [77].

## 3. Autophagy Modulation and Bone Reconstruction

There are three main types of autophagy, namely, macroautophagy, microautophagy, and partner-mediated autophagy [78]. This review focuses on macroautophagy (hereinafter referred to as autophagy), a degradation process during which cellular wastes, such as damaged macromolecules and organelles, are accumulated at lysosomes by autophagy vesicles and removed [78]. Autophagy begins with cytoplasmic organelle isolation in bi-membranous vesicles called autophagosomes, which then fuse with lysosomes to form autophagosomes to degrade its contents by lysosomal hydrolases, such as damaged organelles, intracellular pathogens, glycogens, lipids, and nucleotides proteins, which then turned into a nutrient source for maintaining cellular activity [79]. In this progress, the cytosolic form of microtubule-associated protein 1A/1B-light chain 3 (LC3-I) is converted to form LC3-phosphatidylethanolamine conjugate (LC3-II), which is attached to the autophagosome membrane and then degraded [80]. The transition from LC3-I to LC3-II is considered one of the hallmarks of autophagy. Autophagy plays a quality control role in cell homeostasis [81].

Autophagy is one of the main mechanisms promoting cell survival, which is activated under stress conditions such as nutrient deprivation, oxidative stress, hypoxia, and infection [82]. For example, autophagy promotes the circulation of cellular components, thus providing energy for starving cells [83]. On the other hand, autophagy functions on clearing dysfunctional/damaged proteins and organelles [84]. For example, autophagy mediates the clearance of damaged mitochondria, also known as mitochondrial autophagy, inhibits the accumulation of reactive oxygen species (ROS), thereby protecting cells from oxidative stress and apoptosis [85]. These functions are thought to be essential in bone cell differentiation and immune cell polarization; thus, autophagy is believed to play a central role in bone regeneration (Figure 2).

### 3.1. Autophagy in the Differentiation/Function of Osteoclasts and Osteoblasts

Bone is a metabolically active tissue composed of a network of various types of cells through multiple factors. To maintain physiological bone metabolism, different types of cells (e.g., stromal cells and immune cells) need to continuously interact with each other to ensure osteoblast differentiation, functional mineralization, and osteoclast phagocytosis. The process requires close coordination between cellular organelles and regulators and consumes a large amount of biological energy [86]. Bone metabolic homeostasis is maintained by the balance between osteoblast-derived bone formation and osteoclast-derived bone absorption [87]. Recent studies have confirmed that autophagy is involved in the mineralization process of osteoblasts and the maintenance of bone homeostasis [86]. Autophagy plays an essential role in the differentiation and function of osteoblasts and osteoclasts during bone regeneration. During the receptor activator of nuclear factor-κB ligand (RANKL)-induced osteoclast differentiation, autophagy-associated protein ATG 5/7/12 expression and LC3-II /LC3-I ratio increase with the degradation of p62 [88]. ATG5/7/4B and LC3 have also been reported to play a decisive role in regulating the production of osteoclast wrinkle boundary and lysosome secretion, thus determining the function of osteoclasts in vitro and in vivo [89].

On the other hand, autophagy participates in the differentiation and mineralization of osteoblasts. Autophagosomes act as cargos transporting intracellular mineral crystal-like structures to facilitate extracellular mineralization [90]. Inhibition of autophagy can result in impaired mineralization in vitro and reduced bone mass and volume in vivo, which is followed by oxidative stress and the production of RANKL in general [91]. These results suggest the fundamental role of autophagy in osteoblast differentiation and mineralization, which acts as a mineralization carrier to protect osteoblasts from increased oxidative stress and, in addition, to reduce the production of RANKL, thereby inhibiting osteoclastogenesis during bone formation [91].

### 3.2. Autophagy-Associated Immunomodulation in Bone Remodeling

Not only directly involved in the differentiation and function of osteoblast and osteoclast, autophagy also regulates the immune system which, in turn, regulates bone regeneration through modulating the immune microenvironment [13]. Among the immune cells, macrophages play an important role in the innate immune system. Macrophages are divided into un-activated M0 macrophages, proinflammatory M1 phenotype, and anti-inflammatory M2 phenotype. M1 macrophages are usually activated by microbial lipopolysaccharide (LPS) or Th1 cell-derived IFN, which are considered to promote osteoclastogenesis [92,93]. M2 macrophages, which are usually activated by TH2 cell-derived IL-4 or IL-13, are considered to be the subtypes that inhibit osteoclast differentiation and promote bone regeneration [92,93]. Especially in biomaterial-associated bone regeneration, macrophage phenotype switch from M1 to M2 is considered as an essential strategy in material design/development [94].

Autophagy plays an immunosuppressive role in macrophage inflammatory response. Atg5- or Atg16L1-deficiency on macrophages was found to induce the conversion of M2 macrophages into M1-like phenotypes with enhanced secretion of proinflammatory cytokines [95,96]. Mice with macrophage-specific ATG5-knockout showed induced systemic inflammation [97]. Primary bone marrow-derived macrophages (BMDMs) obtained from this mice type showed abnormal polarization with increased M1 polarization and decreased M2 polarization, indicating that inhibition or deficiency of autophagy can upregulate inflammation in macrophages [97]. Further studies have found that autophagy facilitated the clearance of damaged mitochondria (mitochondrial autophagy, mitophagy). This process can effectively eliminate dysfunctional or damaged mitochondria, which can trigger inflammation and cause cell apoptosis or necrosis, thereby inhibiting inflammation and preventing unnecessary cell loss [98]. Autophagy plays a quality control role in inflammation regulation, and poor quality control can lead to inflammation and cell population death [98]. As previously mentioned, the inflammatory response of macrophages has been shown to induce osteoclast formation and bone loss. At the same time, the transformation of the proinflammatory M1 to anti-inflammatory M2 phenotype is thought to improve bone repair [99]. Therefore, this autophagy-mediated regulation of macrophage response is beneficial to bone regeneration. Nanomaterial-derived autophagy induction has been shown to potentially introduce M2 polarization to improve bone regrowth [100], which further suggests that autophagy may be a potential immunomodulatory target in regenerative medicine, particularly for the treatment of bone loss diseases such as osteoporosis [101], arthritis [102], and periapical lesions [103].

## 4. PNMs Regulate Autophagy in Bone Regeneration

A great number of studies have demonstrated that certain types of PNMs can introduce autophagy activation [104]. PNM-induced autophagy could be a cellular mechanism either against or resulting from nano-toxicity [6]. In addition, both autophagy inhibition [105] and enhancement [106] have been reported as effective bone defect regeneration strategies. From these points, it could be speculated that autophagy acts like a “double-edged sword” in the treatment of bone regeneration. In this review, we summarize the recent progress of porous nanomaterials, including mesoporous silica-based nanoparticles (including mesoporous silica nanoparticles and mesoporous bioactive glass nanoparticles) [8,25,106,107,108], mesoporous hydroxyapatite nanoparticles [109,110,111,112], alumina nanoparticles [10,113], and metal titanium dioxide nanotubes [114,115] in promoting bone regeneration via autophagy modulation (Table 1). We especially emphasize the present mechanisms of autophagy regulation induced by PNMs and the role of autophagy regulation in determining cellular fate.

### 4.1. Mesoporous Silica-Based Nanomaterials (MSNs)

Mesoporous material is one of the most widely used porous materials with pore sizes between 2 and 50 nm [41]. The earliest mesoporous materials developed by researchers were disordered pore structures. In 1992, the scientific research team from the ExxonMobil corporation of the United States [116] first synthesized mesoporous silica with an ordered pore structure by using a surfactant. With the development of synthesis technology, mesoporous silica-based nanomaterials have rapidly become one of the most widely studied nanocarriers [117] due to the fact of their excellent properties including adjustable pores, size and spacing, easy functionalization, and good biocompatibility. Recent advances in biomedicine suggest that MSNs have been extensively investigated and applied in bone regeneration/tissue engineering. Vallet-regi, M., reported the research progress of mesoporous silica nanomaterials in treating complex bone diseases such as bone tumor, infection, and osteoporosis [118]. Chen, L., and Rosenholm, J.M., summarized the application of mesoporous silica nanomaterials in tissue engineering, including bone regeneration, vascular regeneration, and skin regeneration [119,120]. Eivazzadeh-keihan, R., further summarized the research progress of silica nanoparticles in bone tissue engineering [121]. Mesoporous bioactive glass nanomaterials, another kind of porous silica-based nanomaterials, are also widely used in bone regeneration research. Hae-won Kim proved that mesoporous bioactive glass nanoparticles can significantly improve the osteogenic capacity of implant materials, and that by using the mesoporous bioactive glass nanoparticles for surface modification of implants, the surface alkaline phosphate activity and expression of osteogenic genes (ALP, Col, OPN and OCN) [35] could be significantly improved. Bioactive glass modified with cerium oxide has been shown to clear ROS-promoting tissue regeneration and osteogenic differentiation [38,73]. With the extensive research of silica-based nanomaterials, more and more attention has been paid to the mechanism of its regulation on osteogenesis and its potential side effects.

Many studies have shown that silicon dioxide induces autophagy modulation in a variety of cell lines [122]. Ha and co-authors found that silica nanoparticle-associated autophagy stimulation could be the cellular mechanisms of the enhanced osteogenic differentiation and mineralization [25]. They found that silica nanoparticles entered osteoblasts through vesicle-mediated endocytosis, which subsequently stimulated mitogen-activated protein kinase ERK1/2 (P44/P42) [25]. Their findings further revealed that silica nanoparticles stimulated autophagy, including the transition from LC3 MC-I to LC3 MC-II, an essential protein involved in autophagosome formation in an ERK1/2 signaling-dependent manner [25]. In addition, they used a variety of silica nanoparticles with a magnetic cobalt ferrite nucleus (NP1-MNP) to downregulate the relevant proteins and found that LC3 protein and p62, two critical proteins involved in autophagosome formation, bonded directly to silica nanoparticles [25]. Sun, H., et al. synthesized polyethyleneimine-modified porous silica nanoparticles (PPSNs), which can effectively carry plasmid bone morphogenetic protein 2 (PBMP-2), transferred it into MC3T3-E1 cells, and stimulated an autophagy pathway, leading to significant osteogenic differentiation and bone regeneration [107]. Cai, K.Y., et al. compared the effects of silica nanoparticles with different structures on mesenchymal stem cells [8]. It was found that solid silica nanoparticles (with the surface area of 35.938 m^2^/g) had a low capacity of protein absorption and could upregulate the expression of LC3-II through ERK1/2 and v-Akt murine thymoma viral oncogene/the mammalian target of rapamycin (AKT/mTOR) signaling pathways compared with mesoporous silica nanoparticles (with a pore size of 3.5 nm) and biodegradable mesoporous silica nanoparticles (with a pore size of 5.5 nm), thus leading to strong autophagy upregulation in MSCs [8]. The osteogenic differentiation of MSCs with a high level of autophagy was enhanced [8]. Similarly, Sr-doping bioglass nanoparticles provide a promising strategy to promote osteogenic differentiation and bone regeneration in osteoporotic bone defect via early-stage induction of autophagy and later-stage activation of AKT/mTOR signaling pathway [106] in BMSCs.

Blood vessel regrowth is essential for bone regeneration. Sun, Z., et al. reported that negatively charged silicon dioxide nanoparticles resulted in dysregulation of cell homeostasis and angiogenesis through VEGFR2/PI3K/AKT/mTOR and VEGFR2/MAPK/ERK1/2/mTOR signaling pathways [123]. Moreover, there was crosstalk between the VEGFR2-mediated autophagy signaling and angiogenesis signaling pathways. Activation of autophagy by silica nanoparticles inhibited vascularization of endothelial cells [123]. Sun, Z.W., demonstrated that SiNPs triggered autophagy and apoptosis via ROS-mediated MAPK/Bcl-2 and PI3K/AKT/mTOR signaling in endothelial cells and, subsequently, disturbed the endothelial homeostasis and impaired endothelium [124]. Jeong-Ki Min demonstrated that SiO_2_ nanoparticles could induce autophagy-directed cell necrosis via the PI3K/AKT/eNOS signaling axis in a size-dependent manner. The endothelial apoptosis and necrosis induced by 20 nm SiO_2_ nanoparticles were more significant, while the SiO_2_ nanoparticles at larger size had no significant effect [125]. These results may also be a mechanism of cardiovascular toxicity induced by MSNs [123,124,125].

Many studies have found that MSNs can be multicellular cytotoxic. For example, Wang et al. [126] reported that SiNPs increased LC3B expression in hepatocytes in a dose- and time-dependent manner, in accordance with the SiNP-induced cytotoxicity in hepatocytes. The study of Li, M., showed that the cytotoxicity of silica nanoparticles on macrophages was related to surface charge, and the cell survival rate increased with the surface modification of –COOH (negative charge), –NH_2_ (positive charge), and polyethylene glycol (PEG, neutral) [127]. However, some researchers have discovered that MSNs have a positive effect on the cell viabilities of macrophages through inducing autophagy. Low concentrations of silica nanoparticles can protect macrophages from cytotoxicity by promoting an increase in autophagosomes or LC3-II levels [128]. In addition, our previous study reported that silica nanoparticles significantly inhibited the inflammatory response caused by macrophages, possibly due to the activation of autophagy through the Wnt5A/Ca^2+^ pathway [108]. Furthermore, such regulated immune microenvironment enhanced osteogenic differentiation of bone marrow MSCs, leading to increased mineralized nodules and alkaline phosphatase activity [108]. This suggests that the immune regulation mediated by mesoporous silica nanomaterials can be utilized in bone regeneration nanotherapy. Taken together, MSNs, a typical porous nanomaterial type used for bone regeneration, may function as an effective osteoimmunomodulator via autophagy regulation. MSN features, such as structure, dose-dependence, surface charge, and modification chemical groups, have been demonstrated to induce autophagy in BMSCs, an effect benefitting osteogenesis. Furthermore, MSNs, such as silica NPs, could regulate macrophage response via inducing autophagy, hence, generating an immune microenvironment beneficial for bone regeneration [108]. It is, therefore, speculated that MSNs should be designed with suitable chemical/physical properties, including the particle morphology, surface charge, chemical groups, water affinity, and drug loading /releasing efficiency, to induce favorable osteoimmunomodulation via inducing autophagy (Figure 3).

### 4.2. Porous Nano-Hydroxyapatite (nHAP)

Hydroxyapatite (HAP, Ca_10_(OH)_2_(PO_4_)_6_) has chemical properties similar to the inorganic component of the bone matrix. Due to the fact of its biomimetic structure (similar to natural bone tissue), researchers have been extensively investigating its translational potential, especially the synthetic hydroxyapatite as bone substitutes in the field of biomedical applications. Natural bone tissue is a complex organic/inorganic system composed of HAP and type I collagen fibers. HAP accounts for 70% of the bone, while collagen fiber accounts for 20%, and water accounts for approximately 10% of the whole. Because HAP is chemically similar to the inorganic components of the bone matrix, it has a strong affinity to host hard tissue. Due to the enhanced binding of HAP to host tissue, it has a significant advantage in clinical application over other bone substitutes (such as allograft or metal implants). Compared with micron-sized HAP, nano-HAP showed better biological properties as a bone defect implant and repair material, including induced cell activities, osteogenic properties, cell adhesions, and cell–matrix interactions [129,130], which are due to the fact of its peculiar nano-size properties (e.g., high surface area). Furthermore, porous HAP with high porosity and specific surface area has shown excellent biodegradability [131]. Molino, G sumeri [72] summarized the synthesis methods of mesoporous HAP and provided the mechanisms regarding the action of several surfactants (Cetyl trimethyl ammonium bromide (CTAB), F127, Triton X-100 as well as vitamin C) to control the size, shape, and porosity. Specifically, this method is based on using the surfactant as the template. The synthetic raw materials of HA are self-assembled cooperatively around the surfactant micelles as the template, and finally, calcination is performed to remove the templating agent and produce the pores in the particles [72]. As mentioned above, HAP has excellent osteogenic properties, and mesoporous nanometer HAP has a more outstanding performance in bone regeneration. Nga N.K. [132] using synthesized HAP nanoparticles to simulate biological apatite via a hydrothermal method using eggshells as precursors of biological calcium and with the assistance of CTAB. The HAP nanoparticles have a 2–6 nm pore size structure and have good biomineralization and protein adsorption capacity [132]. Frasnelli, M., [133] and Alshemary, A.Z., [134] synthesized strontium-doped mesoporous HAP nanoparticles via water deposition, which significantly promoted the proliferation and osteogenic differentiation of osteosarcoma cells in vitro, and strontium doping could significantly improve the expression of ALP.

Our previous work showed that HAP with different morphologies can activate autophagy of the rat MSCs to promote blood vessel and bone regeneration [111]. The scaffold constructed by nHAP with sphere morphology can also promote osteogenic differentiation by regulating autophagy, which has been further verified in vivo [111]. To investigate the mechanisms behind nHAP-promoted bone regeneration, Tian, W., at al. demonstrated that the internalized nHAP were located in typical autophagy vacuoles along with the increased LC3II/LC3I ratio, indicating that nHAP should be able to induce autophagy [112]. Further studies have found that HAP-induced autophagy is achieved through the mTOR signaling pathway in a concentration-dependent manner [112] which, in turn, enhances the osteogenic differentiation [112]. Another study verified that polydopamine-templated hydroxyapatite (tHA) resulted in induced cell cytotoxicity and reactive oxygen species (ROS) accumulation by inhibiting the expression of autophagy-related proteins beclin1 and LC3II in human periodontal ligament stem cells (hPDLSCs) [109]. The combination of tHA and metformin may prevent cytotoxicity in hPDLSCs exposed to tHA by reducing ROS via regulations on autophagy-related AMPK/mTOR signaling pathways, an effect that enhances the osteogenic differentiation of hPDLSCs [109]. On the other hand, hierarchically constructed bone-mimetic selenium-doped hydroxyapatite nanoparticles (B-SeHANs) promote the induction of autophagy and apoptosis via the ROS-mediated JNK-activation and AKT/mTOR-inhibition pathways in human MNNG/HOS osteosarcoma cells that suppressed bone tumor growth and reduced bone destruction [110]. From the above studies, it can be concluded that (Figure 4) nHAP can affect cell behaviors, such as cytotoxicity, differentiation, bone tumor growth, and bone destruction, by targeting autophagy in different cell types, which indicates that modulating autophagy may be a key mechanism for nHAP to promote bone regeneration.

### 4.3. Titanium Dioxide Nanotubes (TiO_2_ NTs) and Alumina Nanoparticles (Al_2_O_3_)

Metallic oxide nanoparticles are widely used in bone repair and reconstruction due to the fact of their unique and excellent properties. Metallic oxide nanoparticles are being used in bone-related investigations divided into three main categories: bioactive-molecule delivery, cell labeling, and surface modifications of implants and scaffolds. [135]. However, it is unclear about the mechanical and effect of metallic oxide nanomaterials in the skeletal system, especially the effects of these materials on autophagy caused are still largely unclear. This section mainly summarizes the current knowledge regarding porous metallic oxide nanomaterial (including TiO_2_ NTs and Al_2_O_3_ nanoparticles) derived autophagy regulation in bone regeneration.

Recently, considerable attention has been devoted to the generation of titanium dioxide (TiO_2_) through electrochemical anodization, a method that produces nanostructures to improve biocompatibility and cellular behavior on the surface of metal-based implants such as Ti, Ti-based alloys, tantalum (Ta), and zirconium (Zr) [136]. In particular, TiO_2_ NTs with a diameter in the range of 30–100 nm have been found to facilitate cell attachment [63] and osseointegration [64,65]. Awad, N.K., summarized the potential applications of TiO_2_ NTs in clinical implants [66]. Compared with untreated titanium, TiO_2_ NT-modified titanium enhanced the deposition of type I collagen when implanted into a porcine frontal skull [67]. In addition, such implants showed good contacts with bone and would not be damaged by simple stress [67]. In a tibial implant model of rabbits, the TiO_2_ NT-modified implants achieved a nine-fold increase in the bone binding rate compared to the non-modified implants [68]. In vitro and in vivo studies have shown that TiO_2_ NTs can increase the deposition of calcium and phosphorus and enhance the expression of osteogenic factors such as alkaline phosphatase (ALP), osterix (Osx), and collagen-I (COL-I) [69]. Furthermore, functional modifications to the TiO_2_ NT surface, such as combining with metal nanoparticles (gold nanoparticles or silver nanoparticles), grafting peptides, and delivering bone morphogenetic protein 2 (BMP2), can further improve the osteogenic properties of materials [137,138].

However, previous studies have found that the controversial effects of TiO_2_ NPs on cell fate (regarding death and survival). Some studies discovered that TiO_2_ NPs caused cell toxicology, while others reported that the proliferation and survival were enhanced in human fibroblast cells, epithelium, and osteoblasts cultured on TiO_2_ NP-treated titanium surfaces [139]. Wang, H., has shown that autophagy regulates the Wnt/GSK3/β-catenin/cyclin D1 pathway in TiO_2_-stimulated BMSCs to stimulate cell proliferation, autophagy-related protein (LC3II/I) expression in a TiO_2_ NP concentration-dependent manner, which indicates that autophagy may be an essential mechanism for maintaining cell homeostasis under TiO_2_-stimulated cell stress conditions [140]. Our studies used the exosomes from BMP2-stimulated macrophages to intrigue titanium oxide nanotubes to promote bone regeneration [115]. In this work, we found that autophagy could be activated during osteogenic differentiation by the functionalized titanium oxide nanotubes [115].

Porous metallic oxide nanomaterials have also been found to regulate immune response via autophagy modulation. Xiong, W., reported that silver nanoparticle-loaded TiO_2_ nanotubes (Ag@TiO_2_-NTs) could inhibit glucose transport protein type 1 (GLUT1) and promote autophagy levels in macrophages via suppressing phosphoinositide 3-kinase PI3K/AKT pathway, which induced the polarization toward M2-type, a beneficial effect for osteogenesis [114], as the conditioned medium of macrophage activated by Ag@TiO_2_-NTs could significantly promote the expression of osteogenic genes *ALP* and *Runx2* in MC3T3 pre-osteoblasts [114]. In addition, there have been studies regarding the alumina (Al_2_O_3_) nanoparticles with a pore size at the range of 1.3~4.1 nm [141,142]. Inflammation and osteolysis caused by wear debris often lead to implant or joint replacement failure, especially the activation of the NF-κB signaling pathway caused by wear debris has been shown to support the differentiation and maturation of osteoclasts. Zhang, Z., and their team reported that nanosized alumina particles prevented inflammation and osteolysis induced by titanium particles, a common wear debris, via autophagy and NF-κB (RANKL) signaling in MG-63 cells and mouse calvaria osteolysis model [113]. Moreover, Chen reported that Al_2_O_3_ particles promoted fibroblast autophagy in a time- and dose-dependent manner, while RANKL expression (an osteoclastogenesis factor) was negatively correlated with autophagy [10]. The result was further verified in a rat model of femoral head replacement [10]. They further found that LC3II expression was high, whereas RANKL expression was low in patients with the revised total hip arthroplasty with a ceramic interface [10]. This study suggests that autophagy induced by Al_2_O_3_ nanoparticles may have therapeutic potential for the prevention and treatment of osteolysis, which effectively inhibiting osteoclast formation and activation. In summary, TiO_2_ and Al_2_O_3_ nanomaterials can modulate autophagy in MG-63 cells, human fibroblasts, hBMSCs, and macrophage RAW264.7 cells, and autophagy may serve as a key regulator in maintaining cell homeostasis promoting osteogenesis through immune regulation and inhibiting the formation and activation of osteoclasts (Figure 5).

## 5. Challenges and Future Directions

PNMs have been widely used in regenerative medicine due to the fact of their excellent properties such as high porosity and high specific surface area. Autophagy, a process of self-digestion, may serve as an indispensable part of material-directed bone regeneration. Porous bioactive glass was first reported in 2014 to promote osteogenic differentiation by inducing autophagy. This work initialed the study of autophagy generated by bioactive nanomaterials to promote bone regeneration. Different from the previous opinion that autophagy can only promote cell death, this work considered that the induction of autophagy could be conducive to bone regeneration and repair, pointing out an important biological function of PNMs based on regulating autophagy. It provides a new idea for further research and development of bone regeneration materials. Recent studies have further revealed that PNM-associated autophagy modulation is involved in the cell fate and differentiation of both osteoblasts and osteoclasts, thereby influencing the bone remodeling balance to determine bone regeneration fate. Moreover, PNMs have been demonstrated to regulate immune response through autophagy modulation, an effect that should be further investigated to facilitate the ideal osteoimmunomodulation for bone regenerative medicine.

There are still many drawbacks in nanomaterial-derived autophagy regulation, mainly summarized in three aspects. First, at present, the research on autophagy induced by nanomaterials is still shallow and should be further explored. There is a lack of systematic research on the effects of physical and chemical properties of materials, such as properties, structure, morphology, and charge, on autophagy induction of cells, and more systematic and in-depth research is needed. Second, nanomaterials will form a layer of protein crown after entering cells or organisms that will affect the performance of materials in vivo. At present, the regulation of autophagy by the surface protein crown on materials is still lacking. Third, the current research is mostly cell- or animal-based and other preclinical research, and there is still a lack of adequate clinical experimental verifications.

Given the importance of autophagy in bone regeneration, it is necessary to include autophagy modulating function in the design and development of bone regenerative biomaterials. Although material properties, such as morphology, particle size, surface charge/chemical groups, and hydrophilicity have been found to affect cellular autophagy level, it is yet to be explored what are the most ideal features for autophagy up- and downregulation to further refine the material design (Figure 6). In addition, advanced drug delivery system should be developed for specific cell-targeting (e.g., MSC-targeting, macrophage-targeting) controlled-release of autophagy regulators, such as small molecular chemicals (targeting PI3K/AKT, AMPK/mTOR, JNK, and ERK1/2 signaling pathways) and micro RNAs to facilitate different autophagy interventions according to certain cell types, thereby achieving advanced bone regeneration effects. With the considerable development space and good application prospect of nanotechnology in the field of regenerative medicine, PNM-induced autophagy will definitely write a new chapter in tissue engineering in the future.

## Figures and Tables

**Figure 1 pharmaceutics-13-01572-f001:**
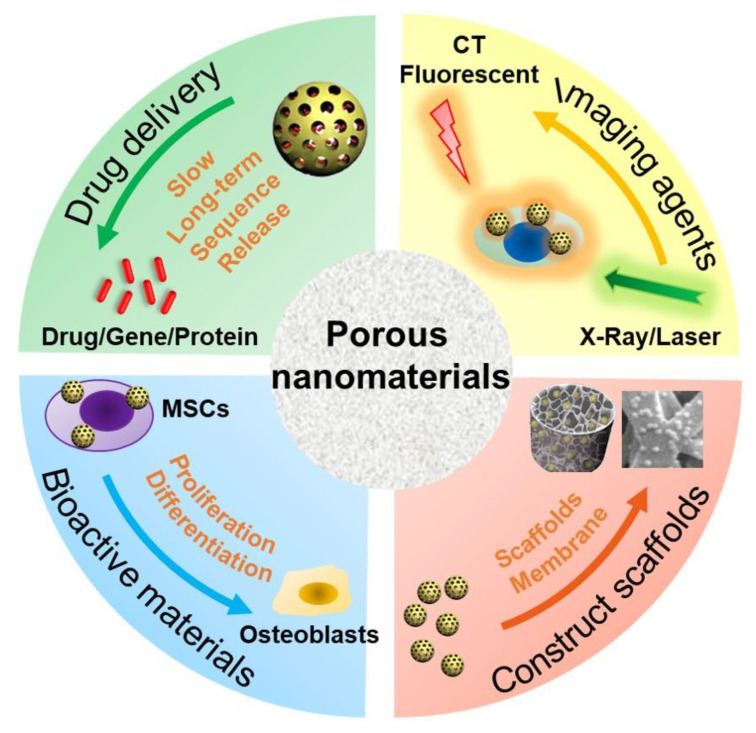
Schematic diagram of PNM applications in bone regeneration. MSCs: mesenchymal stem cells; CT: computer tomography.

**Figure 2 pharmaceutics-13-01572-f002:**
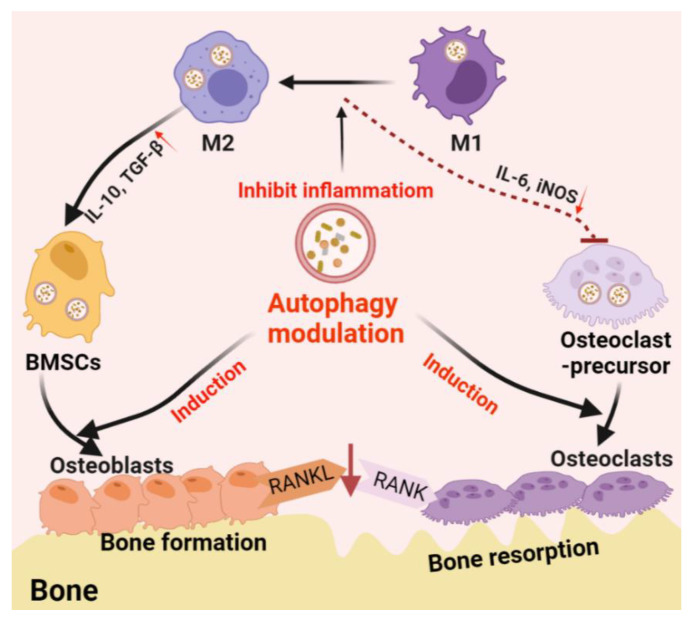
Schematic diagram of autophagy-derived regulation on the differentiation/function of osteoclast/osteoblast and osteoimmunology. On the one hand, autophagy can facilitate the differentiation/function of both osteoclasts and osteoblasts; on the other hand, autophagy induces the phenotype switch from M1 (inflammatory phenotype, which facilitating osteoclastogenesis by producing IL-6 and iNOS) to M2 (tissue-regenerative phenotype, which facilitating osteogenesis by producing IL-10 and TGF-β) in the macrophage population, thereby generating an immune microenvironment favoring bone formation. Furthermore, autophagy induction on osteoblasts can reduce osteoblast-originated RANKL production, hence, reducing osteoclastogenesis by inhibiting the RANKL–RANK signaling pathway.

**Figure 3 pharmaceutics-13-01572-f003:**
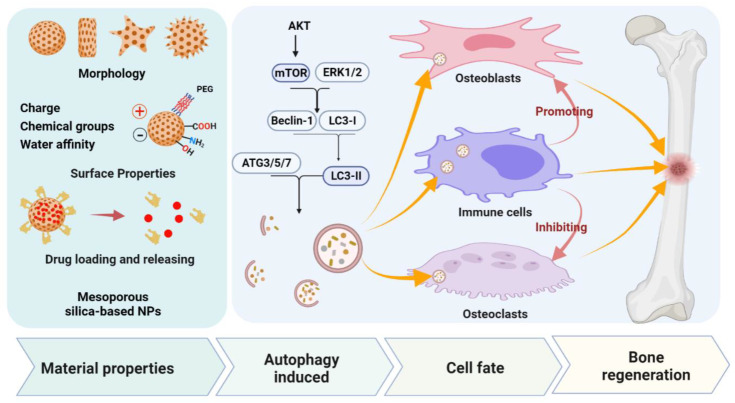
Schematic diagram of MSNs targeting autophagy in bone regeneration. The physicochemical properties of porous silica-based nanomaterials, including morphology, surface properties (charge, hydrophilicity, chemical modification), drug loading, and release, should be the main factors modulating autophagy in bone regeneration.

**Figure 4 pharmaceutics-13-01572-f004:**
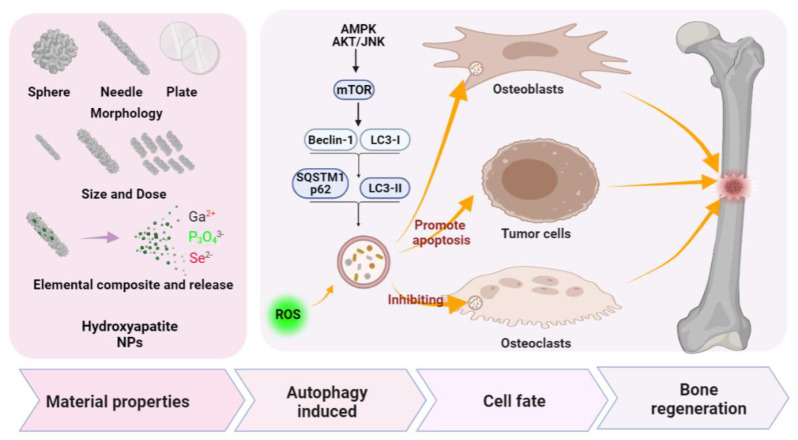
Schematic diagram of hydroxyapatite NPs targeting autophagy in bone regeneration. The properties of hydroxyapatite nanomaterials, including morphology, size, dosage, and added elements, are considered the main autophagy regulators in osteogenesis. (Ga^2+^: calcium ions; P_3_O_4_^3−^: phosphate ion; Se^2−^: selenium ions).

**Figure 5 pharmaceutics-13-01572-f005:**
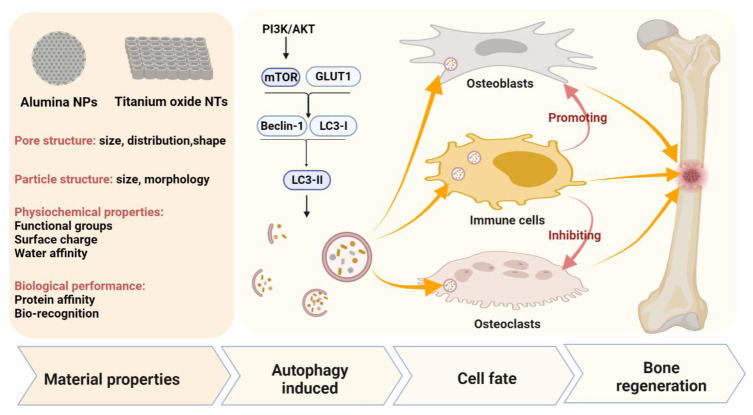
Schematic diagram of alumina nanoparticles and titanium oxide nanotubes targeting autophagy in bone regeneration. The properties of alumina nanoparticles and titanium oxide nanotubes, including pore structure, particle structure, physiochemical properties, and biological performance, serve as the main autophagy modulators in bone regeneration.

**Figure 6 pharmaceutics-13-01572-f006:**
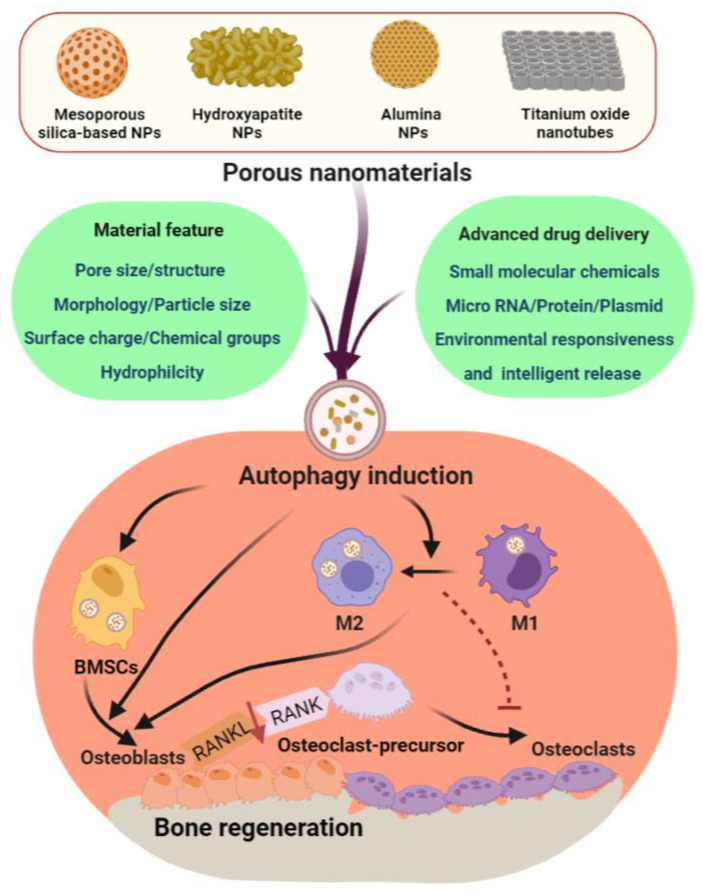
Proposed strategies for future PMN design/development with autophagy-regulation function: one is to further investigate the effects of different material features on cellular autophagy, thereby facilitating the development of optimal autophagy-modulating materials; the other is to develop an advanced drug delivery system to facilitate specific cell-targeting autophagy regulation. Both of these two strategies aim to ensure the selective autophagy modulation on bone marrow stem cells (BMSCs) and macrophages to induce osteogenic differentiation (while reducing the receptor activator of nuclear factor-κB ligand (RANKL) expression and osteoclastogenesis) and macrophage polarization from M1 toM2, hence, generating a microenvironment favorable for bone regeneration.

**Table 1 pharmaceutics-13-01572-t001:** Targeting autophagy using porous nanomaterials: a promising strategy for bone regrowth (↑ stands for up-regulated and ↓ stands for down-regulated).

Item	Nanoparticles	Compound Carried/ Combination Drugs	Target Cells	Autophagy Markers (Down/Up)	Autophagy Mechanism	Osteogenesis Marker (Down/up)	Biological Effect	Reference
1	Silica nanoparticles	Cobalt ferrite magnetic metal core	MC3T3-E1 cells	LC3II/LC3I ↑ P62 ↑	ERK1/2-LC3 and P62	ALP ↑ Alizarin red ↑ OSC ↑	Autophagy and promoted osteoblast differentiation and mineralization	[25]
2	Mesoporous silica nanoparticles	No combination	RAW 264.7 cells	LC3II/LC3I ↑	Not reported	Alizarin Red S ↑	Autophagy and inhibited inflammation and promoted osteogenesis	[108]
3	Silica nanoparticles	Load BMP-2 plasmid	MC3T3-E1	LC3II ↑	Not reported	Alizarin red S ↑	Stimulated autophagy, osteogenic differentiation, and bone regeneration	[107]
4	Silica-based nano-biomaterials	No combination	MSCs	LC3-II ↑ p-ERK/ERK ↑ p-AKT/mAKT ↓ P-mTOR/mTOR ↓	ERK1/2 and AKT/mTOR	ALP, mineralization level, COLI, OPG, OCN, OPN, and RUNX2 ↑	Enhanced the differentiation potential by enhancing autophagy	[8]
5	45S5 bioglass	Sr doped	OVX-BMSCs		AKT/mTOR	ALP, alizarin red S staining ↑	Improved autophagy, promoted osteogenic differentiation of OVX-BMSCs and bone regeneration in osteoporotic bone defects	[106]
6	Nano-hydroxyapatite	No combination	MC3T3-E1 cells	LC3II/LC3I ↑	mTOR	ALP, BMP2, BSP, COL-I, OSC, and Runx2 ↑	Autophagy and modulated osteoblast differentiation	[112]
7	Hydroxyapatite NPs	Integrating nanoparticles within gelatin	rMSCs	LC3A/LC3B ↑ P62 ↑	Not reported	OCN, OPN ↑	Autophagy activation and promoted vascularized and bone regeneration	[111]
8	Selenium-doped hydroxyapatite nanoparticles (B-SeHANs)	No combination	Human MNNG/HOS osteosarcoma cells	LC3B II ↑ Beclin-1 ↑ SQSTM1/P62 ↓	AKT/mTOR and JNK	MMP-9 ↑ bone destruction ↓	Promoted autophagy and apoptosis to inhibit tumor growth while profoundly reducing bone destruction	[110]
9	Polydopamine-templated hydroxyapatite (tHA)	Combined metformin	hPDLSCs	LC3B II ↑ Beclin-1 ↑	AMPK/mTOR	OPN, Runx2, ALP activity, and Alizarin red ↑	THA combined with metformin regulated autophagy, improved the activity of hPDLSCs, and promoted osteogenic differentiation	[109]
10	Nanosized alumina particle	Proteasome inhibitor, bortezomib (BTZ)	MG-63 cells	LC3 ↑	Not reported	Apoptotic cell ↓	Activated autophagy and inhibited apoptosis	[113]
11	Nanosized Al_2_O_3_ particle	No combination	Human fibroblasts	LC3II ↑ Beclin-1↑	BECN-1	RANKL ↓	Autophagy inhibited the expression of RANkL and inhibited osteolysis	[10]
12	Titanium oxide nanotubes	BMP2-stimulated macrophage-derived exosomes	hBMSCs	LC3II/LC3I ↑ ATG5 ↑	Not reported	ALP, BMP2, BMP7, Runx2, OCN, and Col-I, OPN ↑	Activated autophagy during osteogenic differentiation	[115]
13	TiO_2_ nanotubes	Silver nanoparticle loaded	RAW 264.7 and MC3T3-E1	LC3II/LC3I and Beclin-1 ↑	PI3K/AKT and GLUT1	ALP, RUNX2, OCN, and OPG ↑	Activated autophagy and promoted osteogenesis by regulating bone immunity	[114]

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
