# Peer review of "Porous Nanomaterials Targeting Autophagy in Bone Regeneration"

_pharmaceutics, 2021, doi:10.3390/pharmaceutics13101572_

Round 1

Reviewer 1 Report

The review article entitled “Porous nanomaterials targeting autophagy in bone regeneration”. This review paper manuscript described the role of porous nanomaterials (PNB) of small sized (nanometers) in bone tissue regeneration, and as autophagy mechanism in bone remodeling and tissue regeneration. This manuscript is explaining the topic in logical manner and provide significant information on the topic; however, some section and subsection should be extensively edited, and included. I will recommend for major revision. Following comment should be addressed thoroughly.

  1. Although, authors have written lengthy introduction section explaining bone remodeling process, mechanism of autophagy, and PNMs, however, authors missed some porous nanomaterials such as mesoporous bioactive glass (MBGNs), mesoporous ceria (MCeO2), and mesoporous silica coated hydroxyapatite (HA@mSiO2). I recommend authors to highlight these MNMs at Page 2, line no. 83. Some important references are: Acta Biomaterialia 9 (2013) 9508–95219509, J Biomed Mater 2021;109:1457–1467, and ACS Appl. Bio Mater. 2019, 2, 5190−5203, NPG Asia Materials (2014) 6, e90; Biofabrication 11 (2019) 025012, Part. Part. Syst. Charact. 2016, 33, 878–886.
  2. Figure 1, top right “Image agents” should be corrected to “Imaging agents”.
  3. Section 2, “Porous nanomaterials” should be rewrite and updated with new referenced and should also cover the other PNMs mentioned in comment 1.
  4. Section 3, “Autophagy modulation and bone reconstruction” authors should provide schematic diagram to explain the autophagy modulation and role in bone regeneration, which will be helpful for reader to catch-up the topic.
  5. Section 4 should be extensively edited and included subsections with mesoporous bioactive glass, mesoporous ceria, and other mesoporous nanomaterials.

Author Response

Point 1:Although, authors have written lengthy introduction section explaining bone remodeling process, mechanism of autophagy, and PNMs, however, authors missed some porous nanomaterials such as mesoporous bioactive glass (MBGNs), mesoporous ceria (MCeO2), and mesoporous silica coated hydroxyapatite (HA@mSiO2). I recommend authors to highlight these MNMs at Page 2, line no. 83. Some important references are: Acta Biomaterialia 9 (2013) 9508–95219509, J Biomed Mater 2021;109:1457–1467, and ACS Appl. Bio Mater. 2019, 2, 5190−5203, NPG Asia Materials (2014) 6, e90; Biofabrication 11 (2019) 025012, Part. Part. Syst. Charact. 2016, 33, 878–886.

Response 1: We thank the reviewer for this valuable suggestion. Accordingly, we added the contents regarding mesoporous bioactive glass (MBGNs) and mesoporous ceria (MCeO2) into the revised manuscript (Page 2, Lines 83-84, Page 4, Lines 158-159, Page 7, Lines 276-277). The discussion over mesoporous silica coated hydroxyapatite (HA@mSiO2) was incorporated into mesoporous silica nanoparticle section (Page 3, Lines 107-110), as it belongs to the category of composite nanomaterials. All the suggested references (Ref. No. 33, 34, 35, 36, 37, 38) have been cited accordingly (Page 2, Lines 83-84, Page 3, Line 105, Page 4, Lines 141, 158-159,162, Page 7, Lines 303, 304).

Point 2:Figure 1, top right “Image agents” should be corrected to “Imaging agents”.

Response 2: We apologize for the typo and have corrected this mistake accordingly (Fig 1 at the Page 4, Lines 94).

Point 3:Section 2, “Porous nanomaterials” should be rewrite and updated with new referenced and should also cover the other PNMs mentioned in comment 1.

Response 3: We thank the reviewer for this valuable suggestion. In this section, we generally introduced the potential role of porous nanomaterials in osteogenesis. Bioactive glass promoting osteogenesis has been included in this section (Page 4, Lines 139-142, 158-159). We also have added the important references as mentioned (Ref. No. 33, 34, 35, 36). In addition, according to the suggestion, we added the mesoporous ceria (MCeO2) materials as well as related reference (Ref. No. 37, 38, 73, 74, 75) at Page 4, Lines 158-170).

Point 4:Section 3, “Autophagy modulation and bone reconstruction” authors should provide schematic diagram to explain the autophagy modulation and role in bone regeneration, which will be helpful for reader to catch-up the topic.

Response 4: We thank the reviewer for this valuable suggestion.  According to the reviewer suggestion, we provided a schematic diagram (Fig 2 in the revised manuscript) to explain the role of autophagy modulation and bone regeneration (Page 5, Lines 197-198 and Page 6, Lines 199-205).

Point 5:Section 4 should be extensively edited and included subsections with mesoporous bioactive glass, mesoporous ceria, and other mesoporous nanomaterials.

Response 5: We thank the reviewer for this valuable suggestion. Accordingly, we incorporated in contents to include mesoporous bioactive glass and mesoporous silica nanoparticles in silica-based nanomaterials (Page 7, Lines 275-276, 296-303). We summarized the study of bioactive glass regulating autophagy to promote osteogenesis (Page 12, Lines 331-334). In this part, we mainly summarize the autophagy induction properties in bone regeneration, however for mesoporous ceria, we are unable to find its direct regulation on autophagy; thereby we discussed over this material on bone regeneration (Page 4, 158-170).  

Reviewer 2 Report

Reviewer’s comments:

The manuscript entitled ‘Porous Nanomaterials Targeting Autophagy in Bone Regeneration’ has been peer-reviewed. In the present work, the authors have elaborated on the role of porous nanomaterials in bone tissue regeneration events through the autophagy process. The manuscript needs more correction before its possible publication.

Major concerns:

1) The authors need to correct many statements and maintain homogeneity in table 1. Please revise the table. We have pinpointed some words.

  1. a) ALP ↑ (two space indents)
  2. b) &P62 (without space)
  3. c) Silica Nanoparticles (capital N?)

Mesoporoussilica nanoparticles (no space)

silica nanoparticles (starting from small letter)

  1. d) Autophagy,Inhibit inflammation and promote osteogenesis (capital I?)

2) The authors should include the ‘Abbreviations’ section. Many phrases need an expanded form.

  1. a) Fig. 2 PEG, Akt, mTOR,….
  2. b) Fig. 3 Ga2+, P3O43-, Se2-
  3. c) Fig. 5 RANK..

3) Fig. 2-4 needs more explanation in their figure legends.

4) Some molecular species are not represented properly. Check throughout the manuscript.

Ex. TiO2, NH2..

5) AKT or Akt? Please maintain homogeneity for a phrase throughout the manuscript.

6) Fig 1.

  1. a) Provide space between Fig and 1.
  2. b) Correct typo error of ‘Image agents’ in the figure.

Author Response

Point 1:The authors need to correct many statements and maintain homogeneity in table 1. Please revise the table. We have pinpointed some words.

  1. a) ALP ↑ (two space indents)
  2. b) &P62 (without space)
  3. c) Silica Nanoparticles (capital N?)

Mesoporoussilica nanoparticles (no space)

silica nanoparticles (starting from small letter)

  1. d) Autophagy,Inhibit inflammation and promote osteogenesis (capital I?)

Response 1: We apologize for these errors and have thoroughly checked the manuscript to correct them accordingly. All errors in spacing, case and font have been corrected in table 1 (Pages 9-11).

Point 2:The authors should include the ‘Abbreviations’ section. Many phrases need an expanded form.

  1. a) Fig. 2 PEG, Akt, mTOR,….
  2. b) Fig. 3 Ga2+, P3O43-, Se2-…
  3. c) Fig. 5 RANK.

Response 2: We thank the reviewer for this valuable suggestion. As suggested, we added the “Abbreviations” section in revised manuscript(Page 19, Line 574). We also added full names to these abbreviations when they first appeared in the article, including polyethylene glycol (PEG) (Page 12, Lines 353-354), the v-Akt murine thymoma viral oncogene/the mammalian target of rapamycin (AKT/mTOR) (Page 12, Lines 326-327), the Calcium ions (Ga2+), Phosphate ion (P3O43-) and Selenium ions (Se2-) (Page 15, Line 438), and the receptor activator of nuclear factor-κB ligand (RANKL) (Page 6, Line 217).

Point 3:Fig. 2-4 needs more explanation in their figure legends.

Response 3: We thank the reviewer for this valuable suggestion. Accordingly, we added more explanation in the figure legends. We added a new figure (Fig 2.) according to the suggestion of reviewer 1, so the Fig 2-4 were changed into Fig 3-5. The more detailed explanation can be found in the figure legends of Fig 3 (Page 13, Lines 374-378), Fig 4 (Page 15, Lines 435-438), and Fig 5 (Page 17, Line 505-509) in the revised manuscript.

Point 4:Some molecular species are not represented properly. Check throughout the manuscript.

Ex. TiO2, NH2..

Response 4: We thank the reviewer for this valuable suggestion. The molecular species were corrected in the full article including TiO2 (see Page 16, Line 480), -NH2 (see Page 12, Line 353).

Point 5:AKT or Akt? Please maintain homogeneity for a phrase throughout the manuscript.

Response 5: We apologize for the error. We’ve carefully reformatted them into “AKT” in the revised manuscript and corrected all the misuses (Lines 306, 336, 341, 344, 429, 481 and 548).

Point 6:Fig 1.

  1. a) Provide space between Fig and 1.
  2. b) Correct typo error of ‘Image agents’ in the figure.

Response 6: We thank the reviewer for this valuable suggestion. Accordingly, we added space between Fig and 1, and used the "imaging agents" to replace the "image agents" (Page 4, Line 94).

Round 2

Reviewer 1 Report

Authors have edited the manuscript as suggested. I am satisfied with their response. Manuscript can be published.

Reviewer 2 Report

The authors clearly and sufficiently provided replies to some critical issues raised in the previous review stage.

It is considered that the present version of the manuscript was well revised according to all the reviewers' comments.

Thus, this manuscript would be acceptable, unless otherwise decided by other reviewers.

This manuscript is a resubmission of an earlier submission. The following is a list of the peer review reports and author responses from that submission.